# Language-guided Task Adaptation for Imitation Learning

**Prasoon Goyal, Raymond J. Mooney, Scott Niekum**
Department of Computer Science
University of Texas at Austin
{pgoyal,mooney,sniekum}@cs.utexas.edu

## Abstract

We introduce a novel setting, wherein an agent needs to learn a task from a demonstration of a related task with the difference between the tasks communicated in natural language. The proposed setting allows reusing demonstrations from other tasks, by providing low effort language descriptions, and can also be used to provide feedback to correct agent errors, which are both important desiderata for building intelligent agents that assist humans in daily tasks. To enable progress in this proposed setting, we create two benchmarks—Room Rearrangement and Room Navigation—that cover a diverse set of task adaptations. Further, we propose a framework that uses a transformer-based model to reason about the entities in the tasks and their relationships, to learn a policy for the target task.

## 1 Introduction

Imitation learning and instruction-following are two common approaches to communicate a new task to a learning agent, using demonstrations and natural language respectively. However, providing demonstrations for each new task can be burdersome for the user, while providing intricate details using language can also become challenging. This motivates a new paradigm that combines the strengths of both demonstrations and natural language. To this end, we propose a novel setting—given a demonstration of a task (the *source task*), we want an agent to complete a somewhat different task (the *target task*) in a **zero-shot** setting, that is, without access to *any* demonstrations for the target task. The difference between the source task and the target task is communicated using natural language.

For example, consider an environment consisting of objects in a room, as shown in Figure 1. Suppose we have a robot, to which we have already provided the demonstration shown on the left. Now, we want to teach it to go to the opposite side of the table without providing a new demonstration, using a demonstration for the source task and a linguistic description of the difference between the source and the target tasks, such as "Go to the opposite side of the

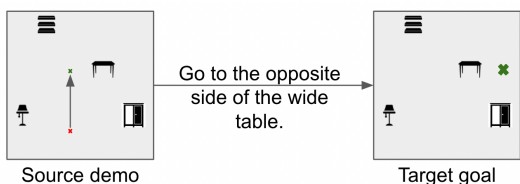

Figure 1: Example of the setting

wide table". Note that, to infer the target goal, neither the source demonstration, nor the description, is sufficient by itself, and the agent must therefore combine information from both the modalities.

This setting (1) allows reusing demonstrations from related tasks, and (2) enables demonstrating complex tasks where both the modalities may be essential. Further, it can be used for correcting the behavior of an agent, where the agent's current behavior can be seen as a demonstration of the source task and a natural language description can be provided to guide the agent towards the correct behavior.

36th Conference on Neural Information Processing Systems (NeurIPS 2022).

## 2 Related Work

Our proposed setting is related to, but distinct from, several prior research directions: (1) instead of getting a demonstration of the desired task as in standard imitation learning [3, 23, 26, 1, 25, 36, 9, 16, 11] and only language in instruction-following [2, 10, 34, 32, 14, 4, 28, 29, 20, 30], in our approach the agent gets a demonstration of a related task, with the difference between the demonstrated task and the desired task communicated using language; (2) our approach can be seen as an instance of transfer learning [31, 35], where the transfer is guided using language; (3) our approach is orthogonal to many other lines of work that use language to aid learning [19, 13, 12, 18, 33, 5, 21].

## 3 Benchmark Datasets

We create two benchmark environments: Room Rearrangement and Room Navigation. The Room Rearrangement Environment consists of a $5 \times 5$ grid, with 2 distinct objects. The goal is to move each object to a desired goal position. The agent and the objects are spawned randomly in the grid. The action space for the agent consists of 7 actions—`Up`, `Down`, `Left`, `Right`, `Grasp`, `Release`, and `Stop`. The Room Navigation Environment consists of a 2D arena, $(x, y) \in [-100, 100]^2$, with 4 distinct objects. The agent is spawned at a random location in the arena, and needs to navigate to a desired goal position. The action space for the agent is $(dx, dy) \in [-1, 1]^2$. We use a common set of objects in both the environments—`Chair`, `Table`, `Sofa`, `Light`, `Shelf`, and `Wardrobe`. Further, each object can have one of 6 attributes—`Large`, `Wide`, `Wooden`, `Metallic`, `Corner`, and `Foldable`.

For each domain, we create three types of adaptations. For Room Rearrangement, these adaptations involve specifying an absolute change in the goal position of each entity, the relative change in the goal position of one entity with respect to the other, and swapping the goal positions of the entities. For Room Navigation, these adaptations involve moving closer to an entity, moving further away from an entity, and going to the opposite side of an entity. For each adaptation template, 5,000 datapoints were generated for training, 100 for validation of the reward and goal learning, 5 for tuning the RL hyperparameters, and 10 for the RL test set. We generate template-based descriptions for all the datapoints. Further, we use Amazon Mechanical Turk to collect natural language paraphrases for 10% of the training datapoints, and all the datapoints in the other splits. More details about the dataset are provided in the appendix.

## 4 RElational Task Adaptation for Imitation with Language (RETAIL)

We propose the RElational Task Adaptation for Imitation with Language (RETAIL) framework that takes in a source demonstration, $\tau_{src}$, and the difference between the source and target tasks described using natural language, $l$, to learn a policy for the target task $\pi_{src}$. The framework consists of two independent approaches, as shown in Figure 2. The first approach—Relational Reward Adaptation—involves inferring a reward function for the target task $R_{tgt}$ using the source demonstration $\tau_{src}$ and language $l$, from which a policy for the target task $\pi_{tgt}$ is learned using RL. The second approach—Relational Policy Adaptation—involves learning a policy for the source task $\pi_{src}$ from the source demonstration $\tau_{src}$, which is then adapted using language $l$ to obtain a policy for the target task $\pi_{tgt}$.

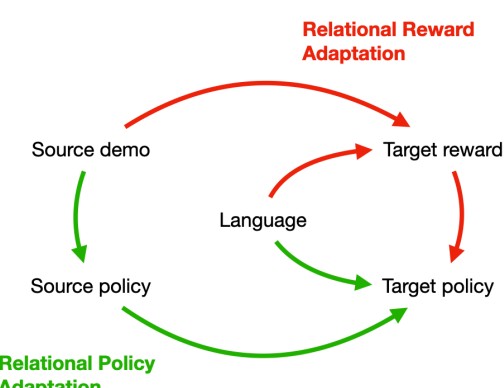

Figure 2: The RETAIL framework

For both these approaches, we assume access to a training set $\mathcal{D} = \{(\tau_{src}^i, \tau_{tgt}^i, l^i)\}_{i=1}^N$, where for the $i^{th}$ datapoint, $\tau_{src}^i$ is a demonstration for the source task, $\tau_{tgt}^i$ is a demonstration for the target task, and $l^i$ is the linguistic description of the difference between the source task and the target task.

We propose a relational model since many adaptations require reasoning about the relation between entities (e.g. "Move the big table two units away from the wooden chair"). Since entity extraction is not the focus of this work, we assume access to a set of entities for each task, where each entity is represented using two one-hot vectors, corresponding to an attribute and a noun. Further, each state is represented as a list, where element $i$ corresponds to the $(x, y)$ coordinates of the $i^{th}$ entity. Finally, we assume that the number of entities, denoted as $N_{entities}$, is fixed for a given domain.

We start by describing some common components used in both the approaches. To encode an entity, its attribute and noun are first encoded using an embedding layer, and the (x, y) position is encoded using a linear layer. These embeddings are concatenated to get the final vector representation of the entity. To encode language, we experiment with 4 encoders: (1) pretrained CLIP model [24], (2) pretrained BERT model [8], (3) BERT model initialized randomly, and (4) GloVE word embeddings [22], with a two-layer bidirectional LSTM [17].

## 4.1 Relational Reward Adaptation

We define the reward $R(s, s')$ using a potential function as, $R(s, s') = \phi(s') - \phi(s)$. Thus, the problem of reward learning is reduced to the problem of learning the potential function $\phi(s)$. We decompose the potential function learning problem into two subproblems: (1) predicting the goal state for the target task given the source goal and the language, $g_{tgt} = Adapt(g_{src}, l)$, and (2) learning a distance function between two states, $d(s, s')$. The potential function for the target task is then defined as $\phi_{tgt}(s|g_{src}, l) = -d(s, Adapt(g_{src}, l))$.

The goal prediction module is a transformer model that takes in the encoded entities for the source goal state and the token embeddings generated by the language encoder, to output the goal state for the target task. The distance function encodes states $s$ and $s'$ using a multi-layered perceptron, and computes an $\ell_2$-distance between the encoded states. The goal prediction module and the distance function are trained independently, and the combined to obtain a reward function that is used to learn a policy for the target task using PPO [27]. More details are provided in the Appendix.

## 4.2 Relational Policy Adaptation

Instead of learning a model to infer the reward function for the target task from the source demonstration and language, in this section, we describe an alternate approach wherein we learn a model to infer the target task policy from the source task policy.

First, a goal-conditioned policy $\pi(s|g)$ is learned using all the source and target demonstrations—given the goal state for a task, $g$, (which is assumed to be the last state in the demonstration), and another state, $s$, we use behavior cloning to learn a policy that predicts the action to be taken at state $s$. We use a neural network to parameterize this policy, wherein the states $g$ and $s$ are concatenated and then passed through a multi-layer perceptron to predict the action at state $s$.

The learned model is then used to generate data of the form (state, language, source action, target action). For each datapoint of the form $(\tau_{src}^i, \tau_{tgt}^i, l^i)$ in the original dataset, the states in the source and target demonstrations are passed through the learned goal-conditioned policy, passing in the source task goal and the target task goal to obtain the actions in the source and target tasks respectively. This data is used to train a transformer-based adaptation model, that takes in the source action, the entities in the state $s$, and the language to predict the target action. See Figure 8 in the appendix for a diagram of the approach.

During evaluation, we are given the source demonstration and language, as before. We use the goal-conditioned policy $\pi(s|g)$ to first predict the action for the current state under the source task, and then pass this predicted action, along with the encoded entities and language to the adaptation model, to obtain the action under the target task. This action is then executed in the environment. The process is repeated until the STOP action is executed or the maximum episode length is reached. Note that this approach does not involve reinforcement learning to learn the policy.

## 4.3 Combining Reward and Policy Adaptation

Recall that the actor-critic model in the PPO algorithm consists of a policy network and a value network. We use the models trained using the policy adaptation and the reward adaptation approaches

to initialize these networks respectively using knowledge distillation [15]. The details of our full approach are described in the appendix.

## 5 Experiments

**Evaluation Metrics.** In the Room Rearrangement domain, an episode is deemed successful if both the entities are in the desired goal locations when the agent executes `Stop`, while for the Room Navigation domain, an episode is deemed successful if

Table 1: Success rates

| Setting | No. of successes | |
| --- | --- | --- |
| | Rearrangement | Navigation |
| Reward Adaptation | $2996.02 \pm 136.21$ | $247.98 \pm 20.51$ |
| Oracle | $4402.78 \pm 410.67$ | $337.22 \pm 7.34$ |
| Zero reward | $121.02 \pm 4.25$ | $0.29 \pm 0.04$ |
| Reward+Policy Adaptation | $8516.78 \pm 894.35$ | $430.80 \pm 5.08$ |

the $\ell_2$-distance between the agent's final position and the desired goal position is less than 5 units. (Recall that the total arena size is $200 \times 200$ units.)

**Relational Reward Adaptation.** We train a policy using the reward function obtained by combining the predicted goal state and distance function for each target task, and report the total number of successful episodes at the end of 500,000 and 100,000 for Rearrangement and Navigation respectively, averaged across 3 RL runs per target task. We compare our approach to two other reward functions: (1) a zero-reward, that gives a zero reward for all actions, serving as a lower bound, and (2) an oracle, that has access to the true goal state for the target task and uses $\ell_1$-distance for Rearrangement, and $\ell_2$-distance for Navigation. Our results, summarized in Table 1 (rows 1-3), show that the proposed approach is about 70% as good as the oracle, and significantly better than the zero-reward lower bound.

**Relational Policy Adaptation.** To evaluate this approach, we generate 100 rollouts using the trained models for each test task, and compute the number of successful episodes. We find that the approach completes 15.33% tasks on the Rearrangement domain, and 3.87% tasks on the Navigation domain. Recall that this approach does not involve RL on the target task.

**Combining Reward and Policy Adaptation.** We report the number of successes when PPO is initialized using the adapted policy in Table 1 (row 4). We observe that on both the domains, initializing the policy network using the Relational Policy Adaptation approach and the value network using the Relational Reward Adaptation approach leads to a substantially faster policy learning on the target tasks, compared to randomly initialized PPO networks. Figure 6 in the appendix shows the learning curves for these experiments.

**Key Takeaways.** To summarize, our experiments demonstrate that: (1) Relational Reward Adaptation leads to successfully learning the target task from the source demonstration and language in many test tasks, but there is room for improvement; (2) Relational Policy Adaptation can be used to complete some target tasks without RL, but there is a significant room for improvement; and (3) combining the two approaches followed by finetuning with RL leads to a much better performance than using either approach independently.

## 6 Conclusions

We introduced a new problem setting, wherein an agent needs to learn a policy for a target task, given the demonstration of a source task and a linguistic description of the difference between the source and the target tasks, and created two relational benchmarks – Room Rearrangement and Room Navigation – for this setting. We presented two relational approaches for the problem setting. The first approach – relational reward adaptation – learns a transformer-based model that predicts the goal state for the target task, and learns a distance function between two states. These trained modules are then combined to obtain a reward function for the target task, which is used to learn a policy using RL. The second approach – relational policy adaptation – learns a transformer-based model that takes in a state, and the action at this state under the source task, to output the action at this state under the target task, conditioned on the source task goal and language. We show that combining these approaches results in effective policy learning on the target tasks.

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

# 7  Appendix

## 7.1  Dataset Details

**Transition Dynamics for the Room Rearrangement Domain.** If the agent is on a cell that contains another object, the `Grasp` action picks up the object, otherwise it leads to no change. A grasped object moves with the agent, until the `Release` action is executed. The `Up`, `Down`, `Left`, and `Right` actions move the agent (and the grasped object, if any) by one unit in the corresponding direction, except when the action would result in the agent going outside the grid, or the two objects on the same grid cell. In these cases, the action doesn't result in any change. The `Stop` action terminates the episode.

**Examples of Adaptations.** Figure 3 shows examples of the adaptations from both the Rearrangement and the Navigation domain, while Table 2 shows some examples of synthetic and natural language descriptions for these adaptations.

Together, these environments cover various types of adaptations, such as specifying modifications to one versus several entities, providing absolute modifications to an entity's position (e.g., "move the table one unit further left") versus modifications that are relative to other entities (e.g., "move the table one unit away from the sofa"). Further, these domains cover different types of MDPs, with Room Rearrangement being a discrete state and action space environment, with a relatively short horizon, while Room Navigation being a continuous state and action space environment, with a longer horizon. (On average, an optimal policy completes a task in the Room Rearrangement domain in about 30 steps, while in the Room Navigation domain in

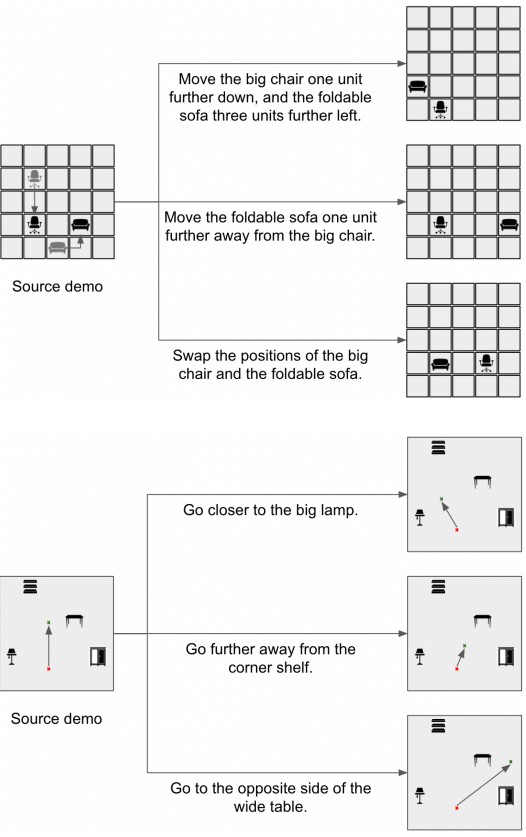

Figure 3: Adaptations used in the Room Rearrangement (top) and Room Navigation (bottom) domains.

about 150 steps.) Finally, the Room Navigation domain has a unique optimal path (i.e. a straight line path between the initial state and the goal state), while the Room Rearrangement domain admits multiple optimal paths (e.g. if reaching an entity requires taking 2 steps to the right and 1 step upwards, these steps can be performed in any order). Thus, these two domains make a robust testbed for developing techniques for the proposed problem setting.

## 7.2  Details of the Relational and Policy Adaptation Approaches

**Goal Prediction.** The goal prediction module is trained by using the final states in the source and target demonstrations, as the source and target goals respectively. We minimize the mean absolute error between the gold target goal state, $g_{tgt}$ and the predicted target goal state, $\hat{g}_{tgt}$:

$$L_{goal} = \frac{1}{N} \sum_{i=1}^{N} \|g_{tgt} - \hat{g}_{tgt}\|_1$$

Figure 7 shows a diagram of the goal prediction module.

**Distance Function.** To train the distance function, two states $s_i$ and $s_j$ are sampled from a demonstration $\tau$, which can be the source or the target demonstration for the task, such that $i < j$. The

Table 2: Examples of template-generated and natural language descriptions collected using AMT.

| | Template | Natural language paraphrase |
|---|---|---|
| 1. | go further away from the metallic table | Increase your distance from the metallic table. |
| 2. | go closer to the foldable light | Move in the direction of the light that is foldable |
| 3. | go to the opposite side of the corner light | Move across from the corner light. |
| 4. | move the large chair one unit farther from the wide couch | Increment the distance of the big chair from the wide couch by one. |
| 5. | move corner table two units further left and metallic shelf one unit further backward | slide the corner table two units left and move the metal shelf a single unit back |
| 6. | move the large table to where the large sofa was moved, and vice versa | swap the place of the table with the sofa |

model is trained to predict distances such that $d(g, s_i) > d(g, s_j)$, where $g$ is the goal state for the demonstration. This is achieved using the following loss function:

$$L_{dist} = - \sum_{s_i, s_j, g} \log \left( \frac{\exp(d(g, s_i))}{\exp(d(g, s_i)) + \exp(d(g, s_j))} \right)$$

This loss function has been shown to be effective at learning functions that satisfy pairwise inequality constraints [7, 6].

The policy adaptation approach is diagrammatically shown in Figure 8.

### 7.3 Details of the combined approach

Here, we describe the details of how the Relational Reward Adaptation and Relationsl Policy Adaptation approaches are combined.

1. Train the reward adaptation and policy adaptation models using supervised learning independently, as detailed in the previous sections.
2. Use knowledge distillation to initialize the value network for PPO, updating the PPO value network towards the potential predicted by the reward adaptation approach.
3. Use knowledge distillation to initialize the policy network for PPO, updating the PPO policy network towards the action probabilities predicted by the policy adaptation approach for the target task.
4. Finetune the action and value networks using PPO with the rewards predicted by the reward adaptation approach.

For knowledge distillation, states from the demonstration data are sampled uniformly at random. Figure 9 shows a diagram of the combined approach.

Importantly, we found that the action network initialized using knowledge distillation usually has a low entropy, and therefore finetuning it directly does not result in good performance. To ameliorate this issue, the entropy of the action network must be kept sufficiently high for it to still allow some exploration. In the continuous control case, we achieve this by increasing the standard deviation of the action network, tuned using the validation set. In the discrete domain, since there is no explicit parameter to control the entropy in the action network, the knowledge distillation step has an additional loss term to penalize low-entropy solutions.

### 7.4 Ablation Experiments for Relational Reward Adaptation

We experimented with several ablations of our reward adaptation model, which we describe below. The results are reported in Table 3, where we include the results from the main paper again in the first 3 rows for easier comparison.

Since our full model consists of two learned components, the goal prediction module, and the distance function, we first study the impact of each of these components independently. We experiment with

Table 3: Success rates for different models on Room Rearrangement and Room Navigation domains. We report both the raw success rates (unnormalized), and success rates normalized by the oracle setting performance.

| | No. of successes | | | |
| Setting | Rearrangement | | Navigation | |
| | Unnormalized | Normalized | Unnormalized | Normalized |
|---|---|---|---|---|
| Reward Adaptation | 2996.02 ± 136.21 | 68.05 ± 3.09 | 247.98 ± 20.51 | 73.54 ± 6.08 |
| Oracle | 4402.78 ± 410.67 | 100.00 ± 9.33 | 337.22 ± 7.34 | 100.00 ± 2.18 |
| Zero reward | 121.02 ± 4.25 | 2.75 ± 0.10 | 0.29 ± 0.04 | 0.09 ± 0.01 |
| True goal, predicted distance | 4164.80 ± 337.83 | 94.59 ± 7.67 | 362.13 ± 12.18 | 107.39 ± 3.61 |
| Predicted goal, true distance | 3706.80 ± 200.46 | 84.19 ± 4.55 | 196.49 ± 12.97 | 58.27 ± 3.85 |
| Synthetic language | 3827.64 ± 141.79 | 86.94 ± 3.22 | 317.11 ± 49.26 | 94.04 ± 14.61 |
| Non-relational goal prediction | 869.89 ± 115.12 | 19.76 ± 2.61 | 0.38 ± 0.17 | 0.11 ± 0.05 |

the following two settings: (1) the true target goal state, with the learned distance function (Row 4), and (2) the learned target goal prediction, with the true distance function (Row 5). As expected, the distance function is easy to learn in these domains, and using the learned distance function instead of the true distance function leads to a small or no drop in performance. Most the performance drop comes from the goal prediction module, and therefore future modeling innovations should focus on improving the goal prediction module.

Next, we look at the performance difference between synthetic and natural language. Row 6 in Table 1 shows the number of successful episodes when using synthetic language only, both during training the goal prediction model, and for learning the target task policy using RL during testing. In both the domains, using synthetic language is significantly better than using natural language, and is comparable to the oracle.

In order to analyze the benefit of using the relational model, we compare our approach against a non-relational model. Row 7 shows the results when using a non-relational model, where we use a multilayered perceptron with three linear layers, that takes in the entity vectors, goal positions of all entities in the source task, and the CLIP embedding of the final token in the description, all concatenated together as a single input vector, and outputs the goal positions of all entities in the target task as a single vector. This model is significantly worse than the relational model on both the domains, highlighting the benefit of using a relational approach for these tasks.

## 7.5 Qualitative Results

In this section, we report some qualitative results on the Navigation domain with reward and policy adaptation approaches.

In Figure 4, we show two examples of goal prediction using the Relational Reward Adaptation approach. In the first example, the predicted goal state is quite close to the true goal state under the target task, suggesting that the model is able to successfully recover the target task. In the second example, the predicted goal is somewhat farther from the true goal. A plausible explanation is that the model was not able to disambiguate the entity being referred to by language, and therefore computes the target goal position as a linear combination of distances to multiple entities.

In Figure 5, we show three examples of paths followed by the agent when following the actions predicted by the Relational Policy Adaptation approach (without any finetuning). In the first example, we see that the agent successfully reaches and stops at the true goal position under the target task. In the other two examples, we see that the agent gets somewhat close to the goal position under the target task, but doesn't actually reach it (and is also going towards the goal position under the source task). The errors seem to get larger as agent gets closer to the target goal, motivating a modified training algorithm wherein datapoints could be weighted differently based on how close the agent is to the goal position. We leave this investigation for future work.

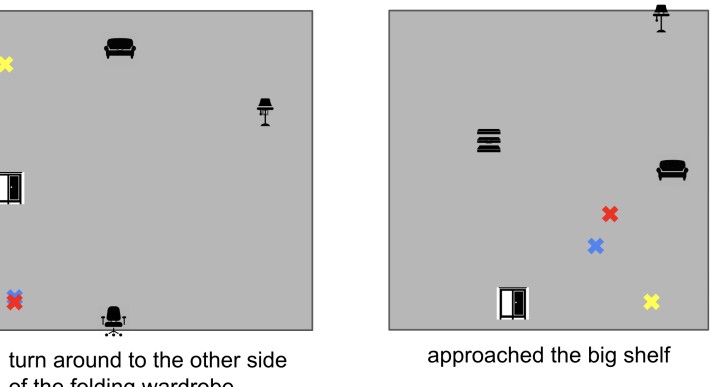

turn around to the other side
of the folding wardrobe

approached the big shelf

Figure 4: Visualization of predicted goal for two test datapoints. The yellow X denotes the goal position under the source task, and the red and blue X's denote the predicted and true goal positions under the target task.

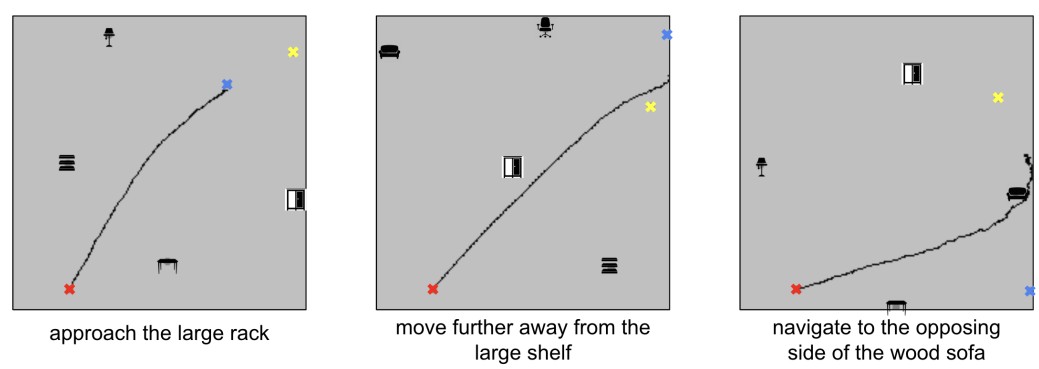

approach the large rack

move further away from the large shelf

navigate to the opposing side of the wood sofa

Figure 5: Visualization of predicted goal for two test datapoints. The red X denotes the initial position of the agent, the yellow X denotes the true goal position under the source task, and the blue X denotes the true goal position under the target task.

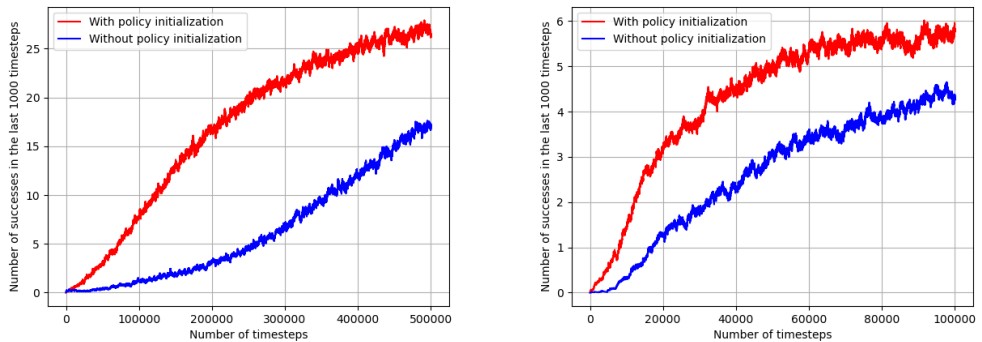

Figure 6: Learning curves comparing the policy training on target tasks when using uninitialized PPO networks and PPO networks initialized using policy adaptation, on the Rearrangement (left) and Navigation (right) domains.

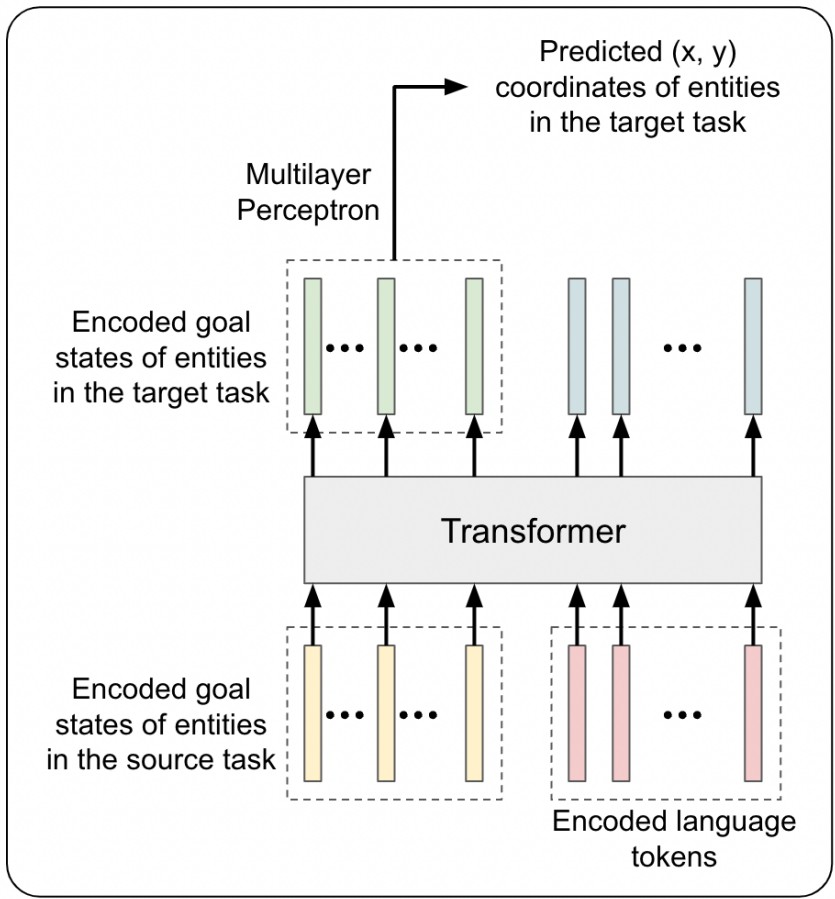

Figure 7: Neural Network architecture for relational goal prediction.

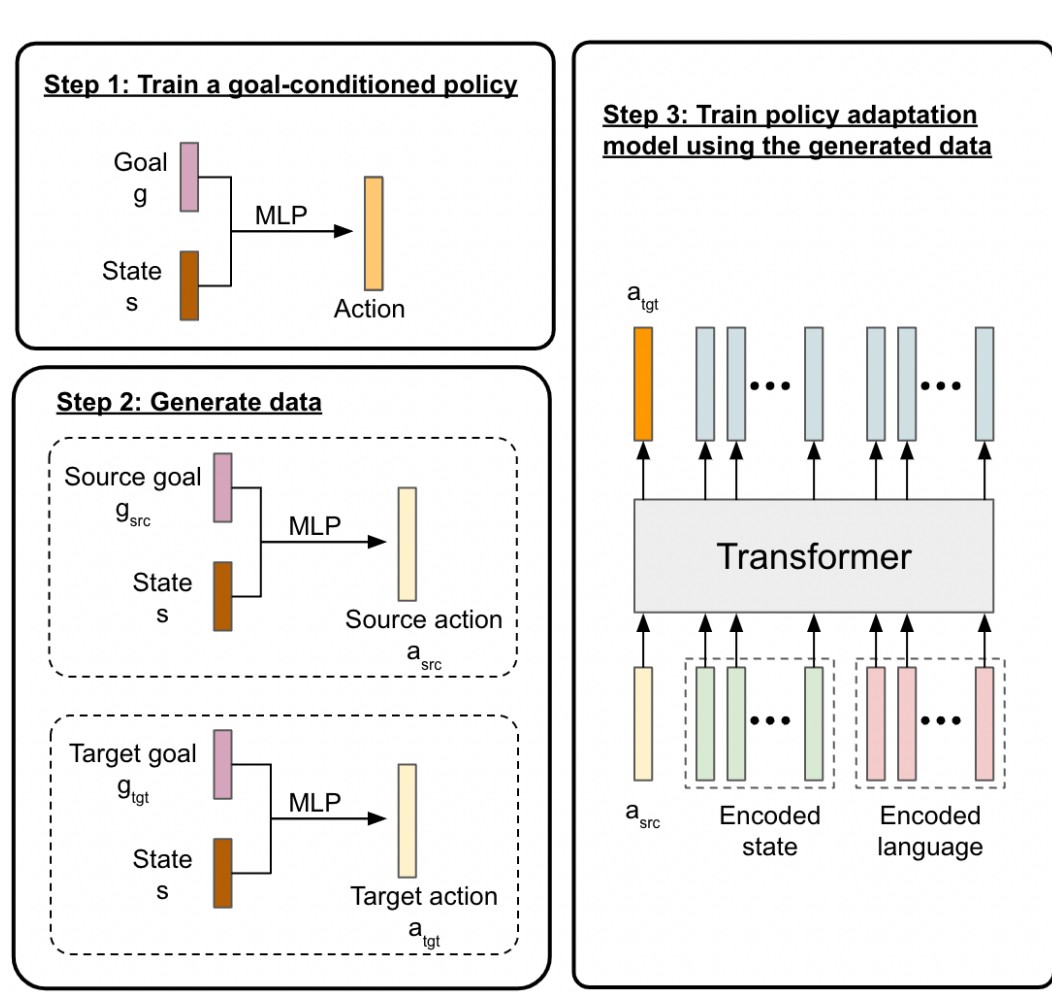

Figure 8: Relational Policy Adaptation approach

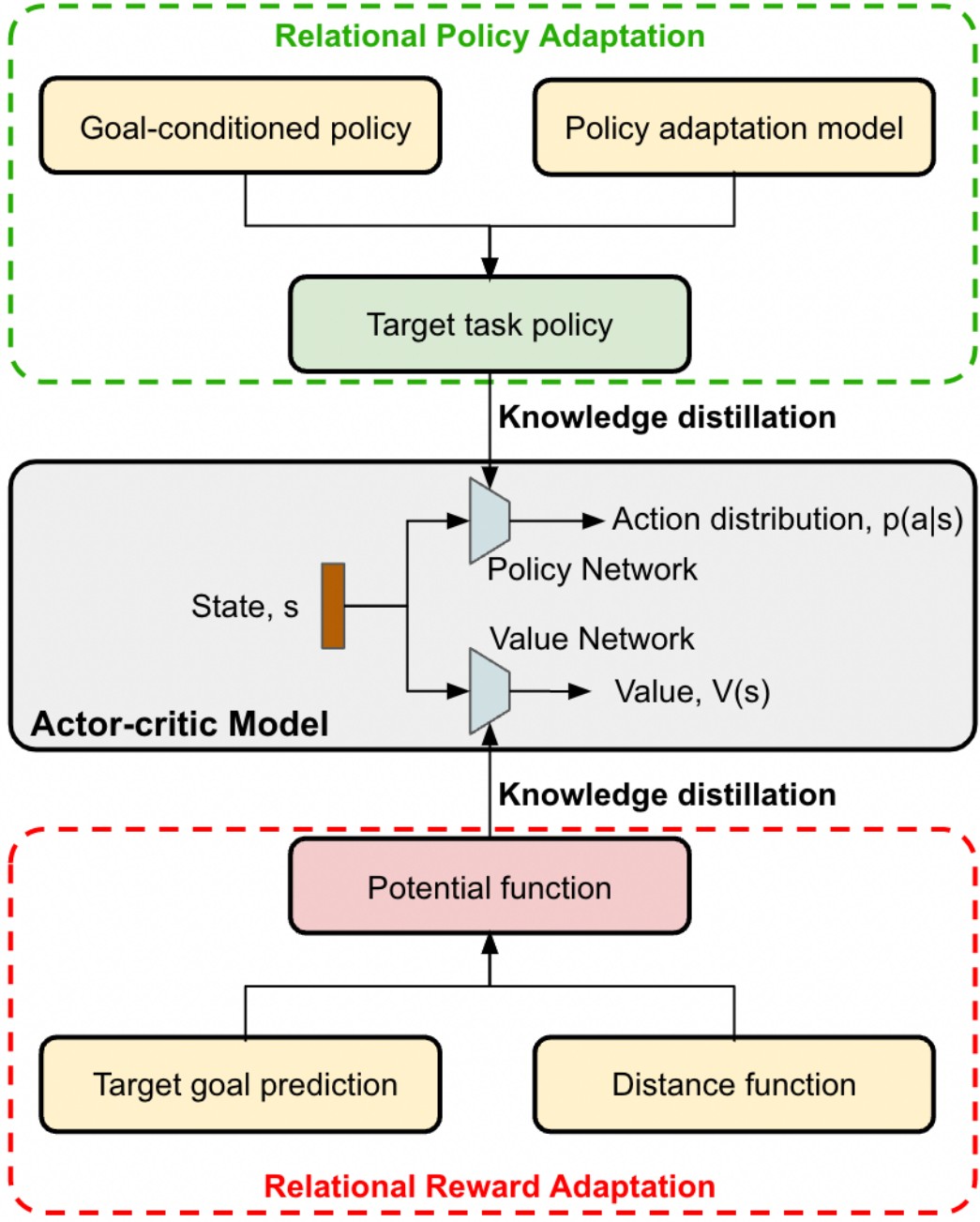

Figure 9: Initializing the value and policy networks of the actor-critic model using the reward adaptation and policy adaptation approaches.

