# OpenReview forum: "Language-guided Task Adaptation for Imitation Learning"
_NeurIPS.cc/2022/Workshop/LaReL — LaReL 2022_

### Official Review · Reviewer_UoYR · 2022-10-09
**Interesting paper but evaluation procedure and results should be made clearer**

**Rating:** 7
**Confidence:** 3

**Review:**

## Summary
This paper proposes a new setting related to imitation learning and instruction following where the agent must learn a target task from a demonstration of a source task and a natural language description of the difference between the source and desired target tasks. To investigate this task adaptation setting the authors propose two benchmarks and the corresponding datasets. The papers presents and implements two independent approaches to tackling this problem (one is more related to Inverse RL while the other is closer to Imitation Learning) and shows how these two can be combined.


## Pros
* proposes a novel problem setting that is well motivated and relevant for real world applications
* provides two benchmarks to investigate the proposed problem setting
* proposes two independent approaches to solving the problem. Both are interesting and well motivated (potential based goal-conditioned reward and learning a goal-conditioned policy to generate a new dataset from which learning a policy)
* proposes a interesting way of combining both methods
* very extensive appendix

## Cons
* the evaluation procedure is unclear to me and it is therefore complicated to assess if the results are encouraging and how much room there is for improvement:
    * Table 1 caption mentions success rates but these are actually No. of successes and it is unclear how to convert those (how many tasks / episodes per task are tested?). From what I understand from l.152, 500'000 and 100'000 episodes are tested so these would make 17% and 0.4% of success rates.
    * l.152 what are the "500'000 and 100'000"? I think a word is missing. Are these new (source demonstration, language description, target goal) datapoints created for evaluation? How are those generated and how do they relate to the training datapoints?
    * Are Relational Reward Adaptation and Relational Policy Adaption evaluate differently? If yes, why is that and why not reporting it in Table 1.?
    * l.159-160: do you consider a task succesful if only one of the 100 rollouts achieves it?
    * If I understand correctly Oracle is training RL with ground truth reward yet it is outperformed by the proposed method? How come, is it because the RL task is already too difficult in itself to be learned and that actually the reward learning part is not that challenging? what about Policy Adaptation + Ground Truth reward? would this be a better upper bound?
    * I understand the problem setting is novel but isn't there some related work that could be adapted to this setting in order to propose some baselines?

## Questions
* For the policy network distillation do you only sample states from the source demonstrations? Otherwise what action do you input to the policy learned by policy adaptation?
* A maybe more straightforward way of combining parts of the reward and policy adaptation would have been to learn a target goal function like in 4.1 (the Adapt(g_src, l) one) and a goal-conditioned policy like in 4.2 ( the pi(s|g) one) and then using this goal-conditioned policy with the adapted target goal. Have you tried it? Do you think it would work/fail and why?

## Typos/Suggestions
* It could be made clearer from the beginning that we have access to target demonstrations for learning (in the introduction and Figure 1 for example)
* Section 3. could be made clearer. What is an adaptation template? Is it swapping the goal position of the entities or is is one instance of doing so? what is a datapoint? is is (tau_src, l, tau_tgt) ? and is there always one of each? i.e. for a given source demo these is only one description and only one target demo?
* what are the natural language paraphrase used for?
* l.71 shouldn't be pi_tgt ?
* l 122. are the states here only the list of coordinates or also the one-hot vectors of attributes? from l. 91 it seems that the state is only the list of coordinates

---

### Official Review · Reviewer_6nFd · 2022-10-18
**Language-guided Task Adaptation for Imitation Learning**

**Rating:** 6
**Confidence:** 5

**Review:**

The paper introduces a novel imitation learning setting, where to perform a new task, the agents see demonstrations from a related task and an easy-to-obtain, natural language signal communicating the difference in the tasks. The agent is then required to execute the new task. Two benchmarks are proposed to investigate this setting.

**Strengths**:
- The proposal of communicating differences in tasks via (easy to obtain) language while being able to reuse past demonstrations is interesting

**Weaknesses**:
- The experimental setting is a bit weak. In the tasks shown in the paper, the goal altogether is being communicated. The idea of communicating the differences in tasks is insufficiently demonstrated.
- In Relational Policy Adaptation: The paper makes an assumption that the demonstration along with the communicated difference sufficiently equips the robot to do the task. What else does the robot need to know in order to "make sense" of the communicated difference? At that point, why is it hard to train a goal-conditioned policy directly? or another way to think about this is how much of a/what kinds of task difference can be adapted to.
- In Relational Goal Adaptation: The paper does not clearly demonstrate why the demonstration is even required to infer the reward for the new task. This relates to the first point listed in the weaknesses.

---

### Decision · Program_Chairs · 2022-10-20

Accept